# Mechanisms Affecting the Biosynthesis and Incorporation Rate of Selenocysteine

**DOI:** 10.3390/molecules26237120

**Published:** 2021-11-25

**Authors:** Jing-Jing Peng, Shi-Yang Yue, Yu-Hui Fang, Xiao-Ling Liu, Cheng-Hua Wang

**Affiliations:** 1Light Industry and Food Engineering College, Guangxi University, Nanning 530004, China; cakejing0520@163.com (J.-J.P.); ysy138481008@gmail.com (S.-Y.Y.); fangyuhui1@outlook.com (Y.-H.F.); 13877173857@163.com (X.-L.L.); 2Guangxi Little Raindrop Technology Co., Ltd., Nanning 530021, China

**Keywords:** selenocysteine, selenoprotein, biosynthesis, affecting factors, mechanism analysis

## Abstract

Selenocysteine (Sec) is the 21st non-standard proteinogenic amino acid. Due to the particularity of the codon encoding Sec, the selenoprotein synthesis needs to be completed by unique mechanisms in specific biological systems. In this paper, the underlying mechanisms for the biosynthesis and incorporation of Sec into selenoprotein were comprehensively reviewed on five aspects: (i) the specific biosynthesis mechanism of Sec and the role of its internal influencing factors (SelA, SelB, SelC, SelD, SPS2 and PSTK); (ii) the elements (SECIS, PSL, SPUR and RF) on mRNA and their functional mechanisms; (iii) the specificity (either translation termination or translation into Sec) of UGA; (iv) the structure–activity relationship and action mechanism of SelA, SelB, SelC and SelD; and (v) the operating mechanism of two key enzyme systems for inorganic selenium source flow before Sec synthesis. Lastly, the size of the translation initiation interval, other action modes of SECIS and effects of REPS (Repetitive Extragenic Palindromic Sequences) that affect the incorporation efficiency of Sec was also discussed to provide scientific basis for the large-scale industrial fermentation for the production of selenoprotein.

## 1. Introduction

At present, there are 22 kinds of amino acids involved in protein biosynthesis. The 21st amino acid [1] (Selenocysteine, Sec) and 22nd amino acid [2] (Pyrrolysine, Pyl) were recently appended to the earlier 20 standard amino acids in the genetic code. Sec and Pyl are nonstandard amino acids encoded by the meaningful termination codon UGA and UAG respectively. This paper focused on the UGA-encoded Sec. Due to the particularity of this codon, selenoprotein composed of Sec inevitably require some special translation and synthesis mechanism in the cell. Studies have shown that the methods to synthesize Sec mainly include chemical synthesis and biosynthesis utilizing animals, plants or microorganisms [3]. Among them, the cost and requirements of chemical synthesis are high, and it is not suitable for large-scale production. The Sec biosynthesis by plants and animals usually require long production cycle but low synthetic efficiency. Comparatively speaking, microbial synthesis of Sec outperforms in low culture cost, mild reaction conditions, considerable fermentation performance and convenient genetic manipulations.

In addition, the selenoprotein synthesized by Sec has a variety of health care and medical functions, and has a broad market application prospect. As one key catalytic amino acid residue of Glutathione peroxidase (GPx), Thioredoxin reductase (TrxR), Selenoprotein P (SELENOP) and some others, Sec involves such physiological activities as anti-aging, anti-inflammation, anti-cancer, detoxification, maintenance of cardiovascular health, improvement of immunity and fertility [4]. These biological functions are mainly carried out by selenases which are the catalytically active form of selenoproteins only by incorporating Sec as the catalytic residue on the active site. In this regard, this paper firstly discussed the pathway of Sec biosynthesis and the influence of its internal components, then pointed out both important factors and elements affecting the incorporation of Sec, and analyzed their action mechanisms afterwards, with a view to promoting the synthetic efficiency of selenoprotein and the subsequent industrial production.

## 2. The Biosynthesis Mechanism of Sec and the Role of Its Internal Influencing Factors

### 2.1. The Synthesis Mechanism of Sec in Prokaryotes and the Role of Main Influencing Factors

#### 2.1.1. Mechanisms of Sec Synthesis in Prokaryotes

At present, the prokaryotic expression systems for the synthesis of Sec are mainly *Escherichia coli* (*E. coli*) [5,6], *Lactobacillus* [7,8] and *Photosynthetic bacteria* (*PSB*) [9,10]. However, genome analysis showed that *Lactobacillus* does not contain selenoprotein-encoding genes and related regulatory factors of selenoprotein biosynthesis, and the molecular mechanism of its biosynthesis and metabolism remains unclear [11]. Moreover, study on the selenoprotein biosynthesis pathway of *Photosynthetic bacteria* is still in its infancy and exploration stage of bacterial strain screening and selenium enrichment condition optimization. Only *E. coli* has a clear translation mechanism which contains a cis-acting element (Selenocysteine insertion sequence, SECIS) and four gene products [12], including SelA (selenocysteine synthetase, SecS), SelB (selenocysteine-specific elongation factor), SelC (tRNA^[Ser]Sec^) and SelD (seleno-phosphate synthase, SPS). They are the important components of the SBIP (Sec Biosynthesis and Insertion Pathway), as shown in Figure 1 [13].

#### 2.1.2. The Role of Four Gene Products in SBIP

Firstly, in prokaryotes, SelC is catalyzed by Seryl-tRNA synthetase (SerS) to load Ser to form Ser-tRNA^[Ser]Sec^. Secondly, as the key conversion compound between inorganic selenium and organic selenium, hydrogen Selenophosphate (H-Se^−^) is the primary source of selenium. After being catalyzed by SelD, low-molecular-weight Selenophosphate (SeP) is synthesized from H-Se^−^ and becomes the most direct selenium donor of Sec [14]. Subsequently, with the participation of SeP, Sec synthase (SelA/SecS) catalyzed the conversion of Ser-tRNA^[Ser]Sec^ to Sec-tRNA^[Ser]Sec^. Finally, another key elongation factor, SelB, recognizes and reversibly binds SECIS and delivers Sec in the form of the previously synthesized Sec-tRNA^[Ser]Sec^ to the A site of the termination codon UGA on the ribosome. A translated quaternary polymer (SelB•GTP•Sec-tRNA^[Ser]Sec^•mRNA) [15,16] would be formed subsequently to achieve the co-translational insertion of Sec into the nascent polypeptide chain, enabling the generation of selenoproteins. In the SBIP pathway, these positive SelA, SelB, SelC and SelD can all promote the expression of Sec, so the reading efficiency of UGA can be improved by appropriately increasing the expression of these four gene products [17].

### 2.2. The Synthesis Mechanism of Sec in Eukaryotes and the Role of PSTK and SBP2 within It

The mechanisms of Sec synthesis in eukaryotes, archaea and prokaryotes are quite similar, except for a few differences. One difference is that the position and structure of SECIS on selenoprotein mRNA are significantly distinct. Another difference is that the translation elongation factor is defined as SelB [18,19] in prokaryotes while eEFSec [19,20] in eukaryotes has slightly different structures. The third difference is that additional steps for two proteases are required [21], such as PSTK (O-phosphoseryl-tRNA^[Ser]Sec^ kinase) and SBP2 (SECIS-binding protein 2), besides SelC, SerS, SelA (SecS), and SelD (SPS2, a kind of selenoprotein [22]). PSTK phosphorylates the hydroxyl on Ser-tRNA^[Ser]Sec^ and converts it to SeP-tRNA^[Ser]Sec^ [19], which is converted to Sec-tRNA^[Ser]Sec^ by SecS afterwards [23,24]. In addition, SBP2, an 846 amino acid protein found in yeast, is another important factor affecting the synthesis of Sec in eukaryotes. After certain folding, SBP2 can interact with eEFSec to promote translation extension, and it also can avoid the termination of translating at UGA. As shown in Figure 2 [1,13], in the eukaryotic Sec expression pathway, SECIS in the downstream untranslated part of mRNA binds first to SBP2, which in turn binds to eEFSec protein. eEFSec presents the recruited Sec-tRNA^[Ser]Sec^ to the UGA codon and makes it bind with the growing amino acid chain to facilitate the synthesis of new eukaryotic selenoprotein.

## 3. Elements on mRNA and Their Functional Mechanisms

### 3.1. The Cis-Acting Element SECIS on mRNA and Its Functional Mechanism

#### 3.1.1. The Functional Mechanism of SECIS in Prokaryotes

##### The Distribution and Introduction Function of SECIS

It has been reported that SECIS contained in prokaryotes are distributed in the Open Reading Frame (ORF) and at the 3’ downstream end of the UGA codon on the Sec-specific selenoprotein mRNA [25,26]. SECIS of *E. coli* formate dehydrogenase F (FDhF) has often been used in prokaryotes as research objects [27,28]. In *E. coli*, SECIS is a cis-acting element to assist Sec insertion, which, in combination with SelB, subsequently promotes UGA translation. As the codon encoding Sec is also the stop codon UGA, its read-through efficiency is very low, only 1–3% of common amino acids, which limits the effective expression of selenoprotein. When the SECIS element was firstly introduced into the ORF of lemon phosphatide hydroperoxidase-glutathione peroxidase (citPHGPx), the successfully expressed mutant citPHGPx was four times more active than the wild-type one without the SECIS [29]. In conclusion, as a signal element, the introduction of SECIS will realize the partitioning between recoding and termination at the stop codon UGA, which is decoded into Sec preferentially. The SECIS introduction accurately and efficiently guides the incorporation of Sec, rather than terminate the synthesis of selenoprotein.

##### The Action of Structure, Sequence and Location of SECIS

As shown in Figure 3 [12], SECIS in prokaryotes is a special “stem-loop” structure [25], and also presents as a highly specific hairpin structure [30], consisting of 40 nucleotides: (1) The secondary “stem-loop” structure (11 nt away from UGA) at the upper end of SECIS is composed of 17 nucleotides between 15 and 31, and the conserved sequence of this structure can be used to predict Sec. Among them, 15–19 and 28–31 nucleotides must be paired to form the stem structure, so as to ensure that U17 is located on the convex ring in the middle of the 5’ end secondary structure. In addition, except for the U17, G23 and U24 sites located on the apical ring, the bases of other sites are highly variable, which provide a basis for the synchronous optimization of its sequences and encoded amino acids, improving the incorporation efficiency of SECIS-guided Sec and improving the properties of selenoprotein. Since U17 and G23 are the binding sites of SECIS and SelB, the mutation of any one of them will lead to the complete loss of UGA readability. (2) In the lower region of SECIS near UGA, the readability is reduced by about 25% because there are unpaired bases between 4–14 and 32–41 [27]. In addition, the difference of the SECIS sequence position in prokaryotic cells corresponds to that of the prokaryotic gene expression process. Since transcription and translation occur simultaneously when prokaryotic genes are expressed, SECIS must be immediately followed by UGA to ensure that the transcription has passed the SECIS sequence when translated into UGA and decoded into Sec.

#### 3.1.2. The Functional Mechanism of SECIS in Eukaryotes 

##### The SECIS’s Distribution and Its Relative Distance from UGA

The SECIS elements on mRNAs for incorporation of Sec into 25 mammalian selenoproteins also exist in eukaryotes and archaea. SECIS elements in eukaryotes and prokaryotes are similar in structure, but the former ones are distributed outside the 3’-untranslated region (3’-UTR) of selenoprotein mRNA [31,32], located far downstream of UGA, and have a relatively large sequence (about 80–150 nt) [33]. Generally speaking, the distance between the codon of SECIS and Sec is at least 51–111 nt to facilitate the translation of the UGA codon [34], while some eukaryotes require a distance of more than 2000 nt to achieve that. However, the far distance of SECIS from the UGA codon will lead to a decrease in the efficiency of its guiding Sec incorporation, then more trans-acting factors are needed to complete the regulation process. The difference of SECIS position in eukaryotic cells also corresponds to that of the eukaryotic gene expression process. The transcription and translation of eukaryotic genes are sequential, so eukaryotes do not have the same SECIS problems found in prokaryotes. SECIS in eukaryotes are located in 3’-UTR, which does not affect the sequence of amino acids in the ORF frame.

##### The Sequence Specificity of SECIS

Eukaryotic SECIS elements were first identified in 5′-DI (deiodinase) and GPx [32], and then two SECIS elements (as shown in Figure 4a [35,36]) were found in 3′-UTR of human selenoprotein P, the only one containing multiple UGA codons [37,38,39]. Between the two ones, SECIS 1 [37,38] plays a major role in selenoprotein synthesis. SECIS have two common conserved sequences among different eukaryotic species, namely the kink-turn structure of the “AUGA:AG” sequences [40,41] in the upper stem part of the inner ring and the “AAR” motif [35] (as shown in Figure 4b [33]). In eukaryotes, the binding protein SBP2 is required to be involved in selenoprotein synthesis, and the kink-turn structure is its anchor site, which is important for UGA recoding. So the middle core area is the primary functional site of the SECIS element. However, the AAR mutation has no effect on the recruitment for eEFSec by the SECIS-SBP2 complex [42]. The conserved region and stem length of SECIS are such important factors affecting the activity of itself that any mutation or deletion of the bases in the conserved region will significantly reduce or even lose the activity of SECIS [43]. When the length of stem on secondary structure is 10–12 base pairs (bp), SECIS maintains high activity [44]. However, some studies have found that the first nucleotide of the kink-turn structure on SECIS of *Caenorhabditis elegans* TrxR is not A but G [45], and mutation analysis showed that A/G was interchangeable.

### 3.2. PSL (Proximal Stem Loop) and SPUR (SelS Positive UGA Recoding) on mRNA and Their Functional Mechanisms

It is widely believed that SECIS is the primary RNA element that controls the insertion of Sec, but recent findings in the lab of Eric M. Cockman [46] suggest that selenoprotein S (SelS) is synthesized differently. The first 91 nucleotides of the 3′-UTR on the protein mRNA contain two conserved regions: (1) PSL: a 34 nt proximal stem loop spanning 3–36 nucleotides. The stem consists of 14 bp and the stem loop consists of 6 nt; (2) SPUR: It belongs to the UGA recoding cis-acting element of SelS and contains an 18 nt nonconserved sequence region of 37–54 nucleotides and a 37 nt conserved region of 55–91 nucleotides located downstream. As shown in Figure 5 [46], only the PSL in the orange box has a highly conserved structure, while the sequence in other parts is variable. It has been reported that only in trials following the alternative insertion of a V5 tag [47,48] does the deletion of PSL increase the incorporation of Sec recoding in SelS-V5 structures, but not when PSL is replaced by other stem loops or non-structural sequences. This indirectly indicates that PSL does not play a positive role in Sec insertion. In addition, the activity of SPUR is independent of the two apical rings, but the readability from UGA to Sec is related to other nucleotide sequences on SPUR. When SPUR mutates at a single point, there is a reduction in UGA recoding, usually by 60% or more. Therefore, SPUR is very important for optimizing UGA recoding.

### 3.3. Translation Competitor RF2 on mRNA and Its Functional Mechanism

RF2 is a release factor (RF) that can read the UGA codon normally and promote the termination of translation. It is also a competitive factor for Sec translation during the expression process. When RF2 is overexpressed, the incorporation efficiency of Sec is only modestly reduced by less than two times, due to the competitive binding to the UGA codon between Sec-tRNA^[Ser]Sec^-SelB-GTP complex and RF2. That is to say, RF2 does not compete with the Sec incorporation mechanism; it only terminates translation on those ribosomes, making Sec incorporation fail in the end [49]. In order to exclude the influence of RF2 on Sec translation, SECIS must be closely followed by the UGA codon. Sec is combined with SelB and GTP to form Sec-tRNA^[Ser]Sec^ -SelB-GTP ternary in the form of Sec-tRNA^[Ser]Sec^, and then inserted into UGA + SECIS preferentially, so that it can avoid the competition from RF2.

## 4. The Specificity of UGA Codon

Almost all selenoprotein genes contain UGA codon encoding Sec (the 21st amino acid) and SECIS element [50]. First of all, UGA is the unique genetic code of Sec in organisms, located in the ORF or 3′-UTR region of selenoprotein mRNA. After being recoded by UGA, Sec would be incorporated and synthesized into selenoproteins. Sequence analysis showed that the ORF of selenoprotein mRNA could contain one or more UGA codons. Moreover, the cDNA transfection results showed that the position of UGA on mRNA was corresponding to the position of Sec in the primary structure of selenoprotein, which confirmed that Sec was absolutely encoded by UGA. Since UGA is also a stop codon, it is necessary for there to be a special translation mechanism to translate UGA into Sec in cells. As mentioned in 3.1 above, the sequential division of the two functions of UGA codon (UGA→Sec and UGA→stop) is realized through the SECIS sequence existing on the mRNA during the translation process, that is, guiding Sec and directly inserting into the polypeptide chain.

## 5. The Structure–Activity Relationship and Action Mechanism of Four SBIP Factors

### 5.1. Structure–Activity Relationship and Action Mechanism of a Synthase Factor—SelA

In early studies of formic acid metabolism in the late 1980s and early 1990s, four selenoprotein-related genes (formerly formic acid dehydrogenase genes) were identified in *E. coli*: SelA, SelB, SelC and SelD, which are specifically responsible for the biosynthesis of Sec and incorporation into selenoprotein. In SBIP, SelA is responsible for the conversion from Ser to Sec on tRNA^[Ser]Sec^. SelA, the synthetase of Sec-tRNA^[Ser]Sec^, has a homodecameric quaternary structure [51], and its internal subunit dimer is similar to the homotetramer SepSecS (Sep-tRNA: Sec-tRNA^[Ser]Sec^ Synthetase) [23,24] in conformation. Both SelA and SepSecS are members of the fold-type-I superprotein family of pyridoxal phosphate (PLP)-dependent enzymes, but their sequences are similar only at the PLP binding site [52,53]. The relative molecular weight (Mr) of SelA is about 500 kDa [51], and the 5′-PLP bound internally is about 50 kDa [54]. In addition to the conserved core and C-terminal region in PLP-dependent enzymes, SelA also has a unique N-terminal region. Karl [54] measured the nucleotide sequence and derived the amino acid sequence of the *E. coli* MC4100 SelA gene early: the initiation codon is ATG, and the termination codon is TGA. Later, Itoh [51] showed that the SelA catalytic site was close to the dimer–dimer interface(as shown in Figure 6 [51]), and the interaction between them was also crucial for the formation of the catalytic site. In summary, all the active sites of the decamer formed by the pentameric reaction of dimers continuously adapt and locate Ser-tRNA^[Ser]Sec^ to synthesize Sec.

### 5.2. The Structure–Activity Relationship and Action Mechanism of a Specific Translation Elongation Factor—SelB

SelB (i.e., Sec-specific translation elongation factor in prokaryotes) is a GTP-binding protein [16,55], belonging to the translation GTPases family (other family members include elongation factors EF-Tu and EF-G, translation initiation factor 2 (eIF2γ/IF2), release factor 3 (RF3) and its eukaryotic homologues). Phylogenetic tree studies have shown that the EFs of Sec in different biological domains are similar in structure, with the N-terminal domains D1, D2 and D3 similar to their EF-Tu corresponding domains and the C-terminal domain D4 responsible for identifying SECIS, while these four domains have slight differences among different species. Therefore, Miljan Simonović [19] used SelB (as shown in Figure 7a [55]), aSelB(as shown in Figure 7b [19]) and eEFSec (as shown in Figure 7c [19]), respectively, for the designation of EF in bacteria, archaea and eukaryotes. Among them, both eEFSec and SelB are homology models based on EF-Tu structure, but there are some differences in several structural domains between them, and the most significant difference lies in D4 (as shown in Figure 7d [19]). For example, D4 of SelB in bacteria consists of four wing-shaped helical folds that rotate around the ligation region, and this structural difference seems to explain why the encoding of Sec in bacteria depends on the neighboring SECIS in ORF without SBP2 involvement.

### 5.3. Structure–Activity Relationship and Action Mechanism of Transporter—SelC

The transporter SelC (i.e., tRNA^[Ser]Sec^) [13,56] is not only found in bacteria, but also in archaea and eukaryotes. The longest tRNA in typical *E. coli* is SelC, which is composed of 95 nucleotides, and its unique cloverleaf secondary structure(as shown in Figure 8 [56]) consists of an 8 bp aminoacyl acceptor stem, a 5 bp TΨC arm, a 20 nt variable arm, a 6 bp Dihydrouridine arm (D arm) and a 4 nt ring. In the primary structure, site 8 corresponds to G, site 14 corresponds to A, sites 10–25 corresponds to a set of Y-R base pairs, and sites 11–24 corresponds to a set of R-Y base pairs. However, SelC lacks Levitt base pairs (G15:C48 or A15:U48) found in other tRNAs in *E. coli*, which are used to maintain the stability of the interaction between the D arm and the TΨC arm. In the SBIP pathway, the first step of the synthesis of Sec is to aminoacylate and synthesize the transporter SelC into Ser-tRNA^[Ser]Sec^ under the action of SerS; after the conversion from Ser to Sec on tRNA^[Ser]Sec^, Sec-tRNA^[Ser]Sec^ is finally gathered in the translation complex SelB•GTP•Sec-tRNA^[Ser]Sec^•mRNA. Therefore, SelC is also known as the UGA decoder, by which Sec is incorporated into the small pieces of selenoproteins.

### 5.4. Structure–Activity Relationship and Action Mechanism of Synthase Factor—Binding Protein SelD

Selenophosphate synthetase (SPS), originally derived from the SelD gene product of *E. coli*, has been confirmed to be present in most organisms [57]. It is a common key enzyme involved in different selenium metabolism processes and belongs to a protein superfamily that contains an ATP-binding domain. There are two types of SPS in mammals: SPS1 and SPS2 [22,57,58]. SPS1 loads Thr substitute at Cys17 site of EcSPS (*E. coli* SPS). SPS2 in the same position is itself a selenoprotein containing Sec, which is responsible for transferring selenium element to selenophosphate and serves as a direct selenium donor for selenoprotein biosynthesis [59]. SPS2 is mainly responsible for catalyzing 5’-adenosine triphosphate (ATP) and hydrogen selenide (HSe^−^) to produce SeP (H_2_SePO_3_^−^), where the reaction formula [57] is as following:$${\mathrm{HSe}}^{-}\;+\;H_{2}O\;+\;\mathrm{ATP}\;\overset{SPS2}{\to }\;H_{2}{{\mathrm{SePO}}_{3}}^{-}\;+\;\mathrm{Pi}\;+\;\mathrm{AMP}$$

Therefore, the synthesis of SeP and selenoprotein can be increased by increasing the synthesis pathway of HSe^−^. SeP is a selenium donor necessary for the synthesis of important organic selenium compounds such as Sec, SeU (selenides present in some tRNA swing sites) and Se cofactor (existed in some molybdenum-containing enzymes). As shown in Figure 9 [58], the SPS subunit consists of two parts: the N-terminal domain (1–156 AA) and the C-terminal domain (157–336 AA). A steep, 30-amino acid channel is formed between them, which appears to be the binding/catalytic site for the substrate. It is speculated that the key catalytic residues Sec/Cys13 and Lys16 should be located on this mobile segment.

## 6. Operating Mechanism of Two Key Enzyme Systems for Inorganic Selenium Source Flow before Sec Synthesis

### 6.1. Operating Mechanism of Transporter System

Since the chemical properties of the selenium element are very similar to that of sulphur, the transport system of sulphur can also be used to transport selenium by microorganisms. It has been reported that selenate (SeO_4_^2−^) can be transferred into cells by the sulfate ABC transport permeability enzyme system of *E. coli* (Cys AWTP) [60]. However, the pathway system of selenite (SeO_3_^2−^) into bacterial cells also includes Gut S, Smo K, Ded A and other transporters [61,62,63]. For fungi, such as *Saccharomyces cerevisiae*, the mechanism of selenium salt transport is similar to that of *E. coli*, and both sulfate permeability enzyme (Sul1 and Sul2) and sulfate transporters (Sul1p and Sul2p) in the system are related to the transport of SeO_3_^2−^ [64]. However, when different concentrations of phosphate (PO_4_^3−^) are added to the culture medium of yeast cells, the transport of SeO_3_^2−^ is constrained by the different affinity of phosphate transporters (Pho84p, Pho87p, Pho89p, Pho90p and Pho91p) [65]. In addition to sulfate and phosphate transport systems, MC Dermott additionally found that the monocarboxylic acid homotropic transporter (Jen1p) in yeast cells is also involved in the transport of SeO_3_^2−^ [66].

### 6.2. Operating Mechanism of Reductase System

Most bacteria cultured in a selenium-rich salt medium have either a nonspecific selenite reductase system (consisting of nitrite reductase, sulfite reductase and glutathione reductase) or specific sulfate reductase, selenate reductase, and fumarate reductase system in the cell periplasm [67]. Some microorganisms in the absence of oxygen, respiration and reduction of two pathways proceed simultaneously. Generally, inorganic selenium sources (SeO_4_^2−^ and SeO_3_^2−^) will eventually be reduced to elemental selenium (Se^0^); for instance, in *Thauera selenatis* (*T. selenatis*) nitrite reductase action, SeO_3_^2−^ will be reduced to Se^0^. However, some microorganisms (such as yeast) are eventually reduced to hydrogen selenide (a key precursor for the synthesis of selenoprotein and selenopolysaccharide) by Se-enriched culture in the presence of sulfate reductase [68,69,70]. In addition, under the action of the selenate reductase (Srd ABC) [71,72] of Gram-positive *Bacillus* selenatarsenatis SF-1 and the selenate reductase (SerABC) [73] of Gram-negative bacteria *T. selenatis* AX^T^, SeO_4_^2−^ in the medium can be generally reduced to red nano-selenium, and these particles generally aggregate in the cytoplasm, cell membrane or medium.

The final products of the above Se-enriched transformation by microorganisms, whether nano-selenium or elemental Se, are insoluble and low-toxicity substances, so this transformation should be controlled as far as possible. However, hydrogen selenide (H_2_Se) is a volatile and highly toxic substance, which must be transformed into other selenium donors (such as SeP) as soon as possible to avoid the effect on the microorganisms themselves. In addition, there is another explanation for selenite’s toxicity: the most abundant and simple glutathione (GSH) found in eukaryotic cells, cyanobacteria and proteobacteria α, β and γ is a mercaptol (RSH) [74]. The selenite (SeO_3_^2−^) in the culture medium can react with the sulfhydryl group on the molecule to form glutathione selenotrisulfide (GS-Se-SG) and release reactive oxygen species O_2_^−^. The superoxide anion O_2_^−^ produced by this process is the source of toxicity [75]. However, the degradation of GS-Se-SG by oxidative kinase can remove this interference and facilitate the further reduction of GS-Se-SG to other selenium donors.

## 7. Conclusions and Discussion

### 7.1. Some Other mRNA Elements Affecting the Incorporation of Sec

The above major mechanisms affecting the incorporation of Sec mainly focus on the key factors such as SelA–SelD and SECIS, which lead to the traditional biosynthesis of selenoprotein. However, in SBIP, there are three additional influences: (1) the size of the translation initiation interval—the optimal mRNA translation initiation interval helps the ribosome to approach the initiation codon, thus accelerating the loading of ribosomes. For example, if there is not enough time for SECIS to refold between successive ribosomal channels, the activation efficiency of FDhF translation of selenoprotein gene is only 40% [49]. (2) Other action modes of SECIS: some SECIS on selenoprotein mRNA only have the “CCC” but not “AAR” motif, so they do not act directly on ribosomes but on factors that change together with this element [35]. If UGA codon shutdown occurs when SECIS is introduced into the mRNA entrance of ribosomes, the adverse effects such as mRNA cleavage and reverse translation may be stimulated, thus affecting the forward translation and Sec incorporation rate [49,76]. (3) Effects of REPS (Repetitive Extragenic Palindromic Sequences): REPS exists in *E. coli*, located at about 500 nt downstream of the stop codon on mRNA, which leads to a large number of stem loop structures in the downstream region. When the distance between REPS and the stop codon is less than 15 nt, the translation is down-regulated; while the distance between the REPS and the stop codon is prolonged, the translation is up-regulated [76].

### 7.2. The Effect of Some Inherent Factors on Selenoprotein Expression

First, foreign elements or foreign proteins may affect the expression of selenoprotein. Peter [77] pointed out that the heterologous selenoprotein SelB from *Bacillus* could not interact properly with the ribosomes of *E. coli* due to the influence of the ribosome fidelity control. Therefore, heterologous SelB can reduce the normal reading of UGA in *E. coli*. However, heterogenous SelC genes do not inhibit the translation of UGA, because tRNA^[Ser]Sec^ genes from many organisms can supplement SelC damage in *E. coli* mutants, so that SelC does not become a bottleneck for Sec-tRNA^[Ser]Sec^ biosynthesis and heterogenous Sec expression [78]. Secondly, inherent factors such as ribosomes affect the read through rate of Sec. Some ribosomes accumulate upstream of the UGA codon [79], causing some ribosomes to terminate translation at UGA rather than binding Sec. Furthermore, the selected plasmid is also significant. For example, the pSUABC plasmid used to co-express the SelA, SelB and SelC genes could increase the yield of recombinant selenoprotein by at least 5–7 times. Finally, other inherent matching problems, such as inactive collation of Sec to UGA, inefficient placement of Sec-tRNA^[Ser]Sec^ or poor reactivity of Sec in peptidyltransferase reactions, may result in a low delivery rate of Sec-tRNA^[Ser]Sec^ to ribosomes [49].

### 7.3. The Exploration and Thinking of New Mechanism

People have formed an inherent impression on the conventional biosynthesis of selenoprotein. To explore the decoding efficiency of UGA, they mainly focus on several key factors in SBIP: one synthase SelA, one transporter SelC, and two binding proteins (GTP binding protein—SelB and ATP binding protein—SelD). However, new methods and mechanisms to promote Sec incorporation have been discovered: in addition to the inherent RF and novel influencing factors (such as PSL and SPUR), Elias S.J. Arnér’s [80] team from Sweden also found that Sec can also effectively incorporate a predefined UAG stop-codon to compete with RF1 (but not RF2). The mammalian selenium protein TrxR with high purity, yield and activity can be produced by using pABC2-rTRS_UAG_ in *E.coli C321.*∆*A* with RF1 depleted. This study opens up a novel method for recombinant selenoproteins with SelB-mediated Sec to be directly inserted into the UAG codon (rather than the traditional UGA). The method still employs the catalytic tetrad of SelB in selenoprotein biosynthesis, but does not rely on SECIS. Human glutathione peroxidase 1 (GPx1) can also be produced with this new system. In short, these new modes of Sec synthesis are still to be discussed and deeply explored.

## Figures and Tables

**Figure 1 molecules-26-07120-f001:**
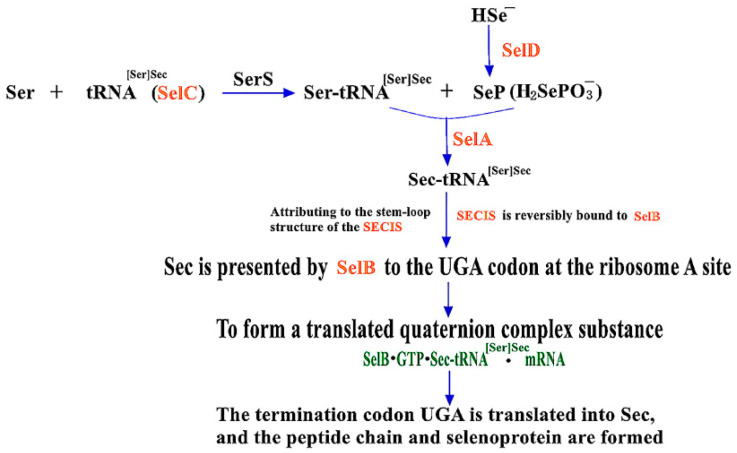
Prokaryotic biosynthesis and insertion pathways of Sec. Adapted with permission from reference [13].

**Figure 2 molecules-26-07120-f002:**
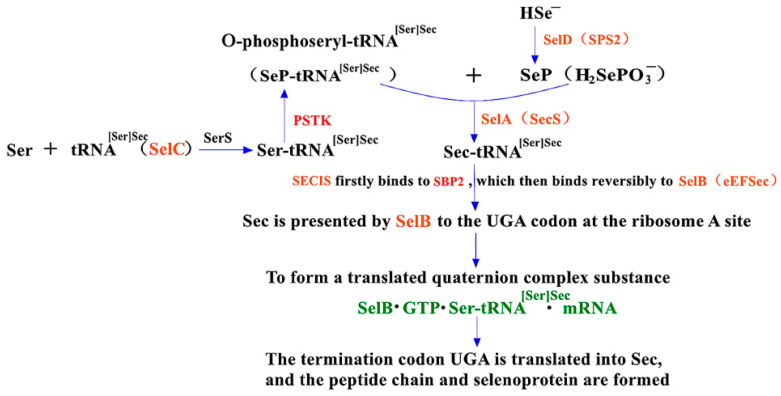
Eukaryotic biosynthesis and insertion pathways of Sec. Adapted with permission from references [1,13].

**Figure 3 molecules-26-07120-f003:**
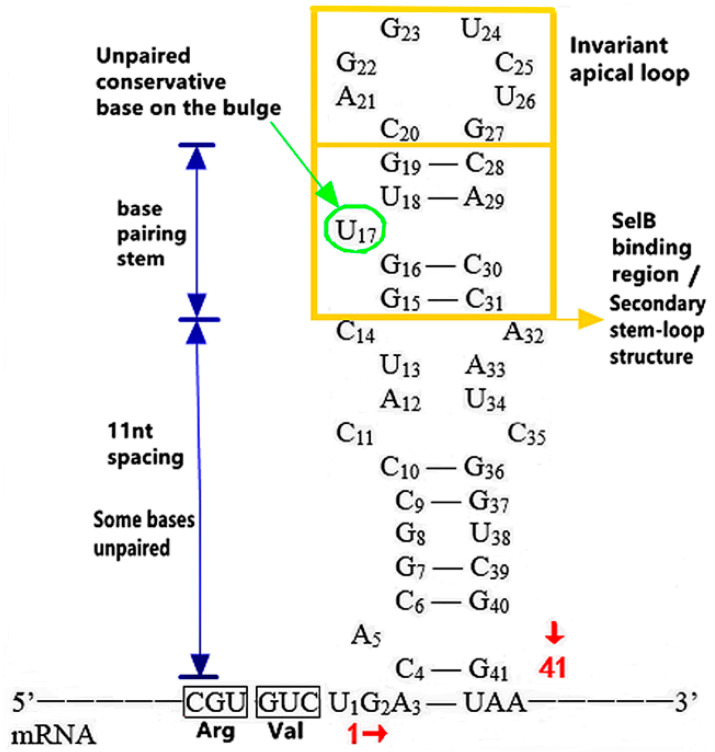
SECIS’s unique sequence structure in prokaryotes. Adapted with permission from reference [12].

**Figure 4 molecules-26-07120-f004:**
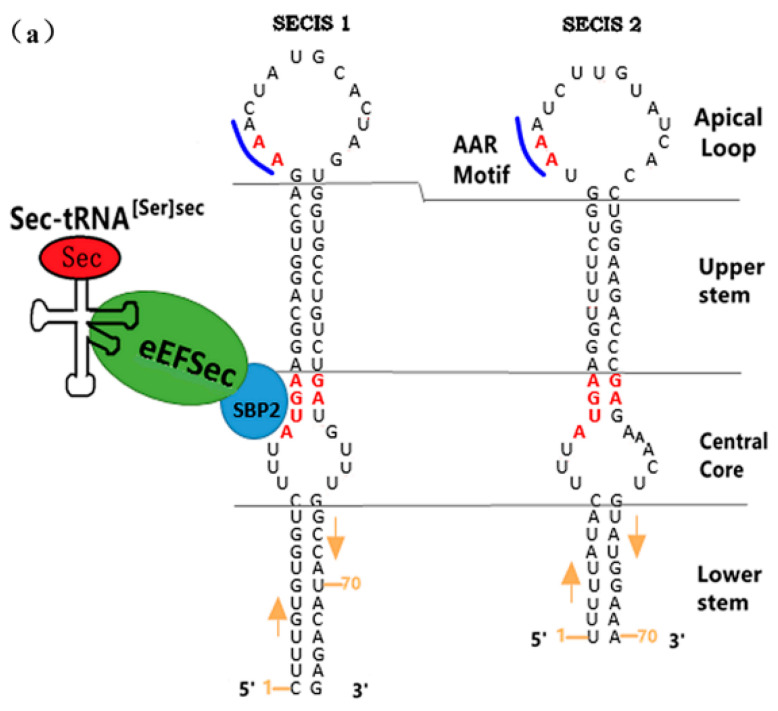
Two SECIS sequences and model structures in eukaryotes. Adapted with permission from references [33,35,36]. (**a**) shows the sequence structures of SECIS 1 and SECIS 2 on human selenoprotein P, including apical ring, upper stem, inner ring and lower stem. The conserved sequences are highlighted in red fonts; the binding sites of SBP2 protein to SECIS and eEFSec are, respectively, highlighted in blue and green circles. (**b**) shows two molecular models of SECIS in eukaryotes. The letter “N” in the middle convex ring between Helix I and II represents characteristic nucleotide. Form 2 differs from form 1 in the additional small ring, which will be formed to stabilize the SECIS‘s secondary structure when the top ring is large.

**Figure 5 molecules-26-07120-f005:**
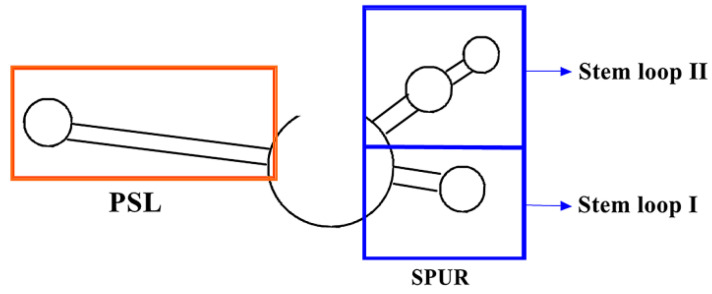
Model structure of PSL and SPUR in 3′-UTR of SelS. Adapted with permission from reference [46].

**Figure 6 molecules-26-07120-f006:**
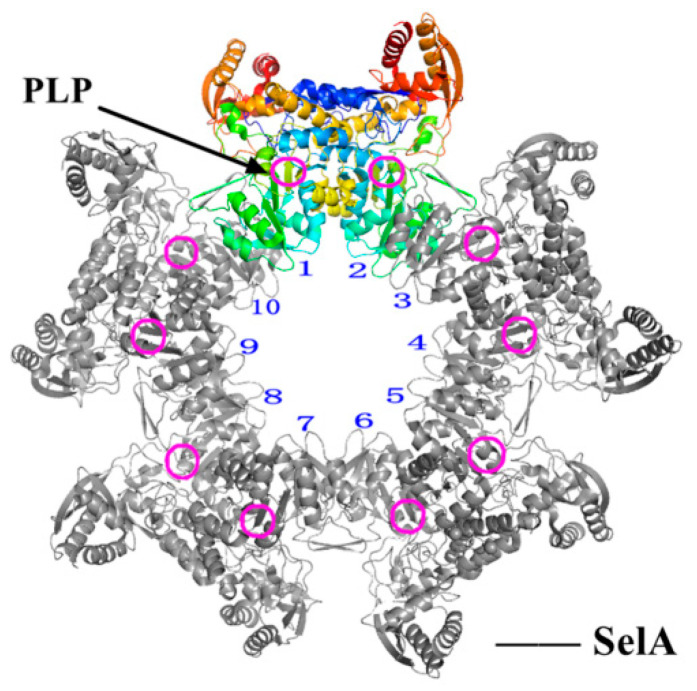
The homodecamer ring structure of SelA. Adapted with permission from reference [51]. This structure is also pentameer quaternary structure of a dimer. Ten subunits are represented as 1~10, in which the nearest dimer 1 and 2, 9 and 10 have the largest spacing, and have been judged as the main catalytic site. And ten PLP binding sites are highlighted as rose red circles.

**Figure 7 molecules-26-07120-f007:**
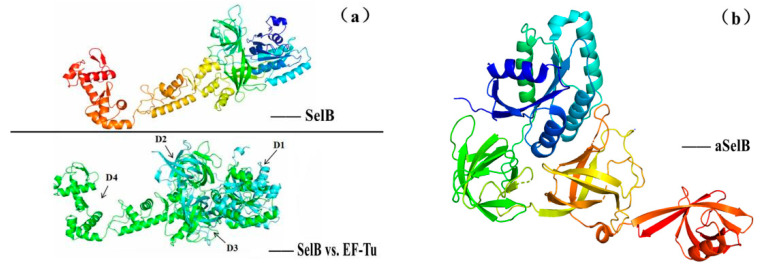
The structure of SelB. Adapted with permission from references [19,55]. (**a**) is overall structure of the full-length bacterial SelB with the similar domains of EF-Tu; (**b**) is the structure of archaea aSelB; (**c**) is the structure of human eEFSec; (**d**) is D4 domain of SelB and eEFSec.

**Figure 8 molecules-26-07120-f008:**
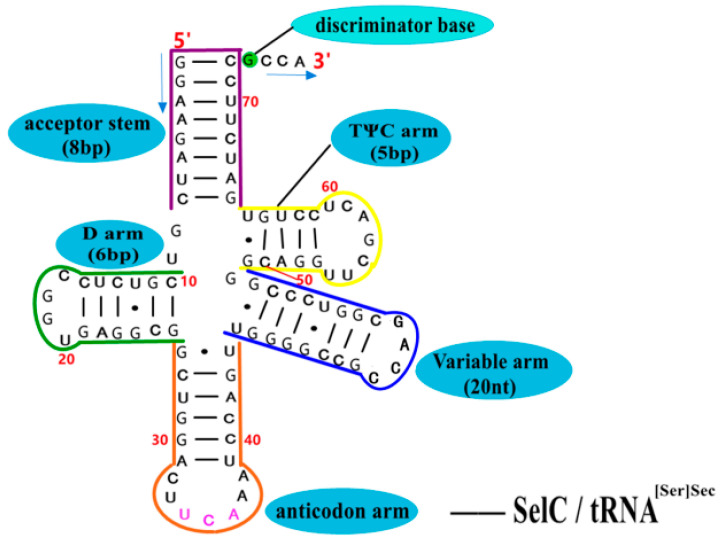
The unique cloverleaf secondary structure of SelC in *E. coli*. Adapted with permission from reference [56].

**Figure 9 molecules-26-07120-f009:**
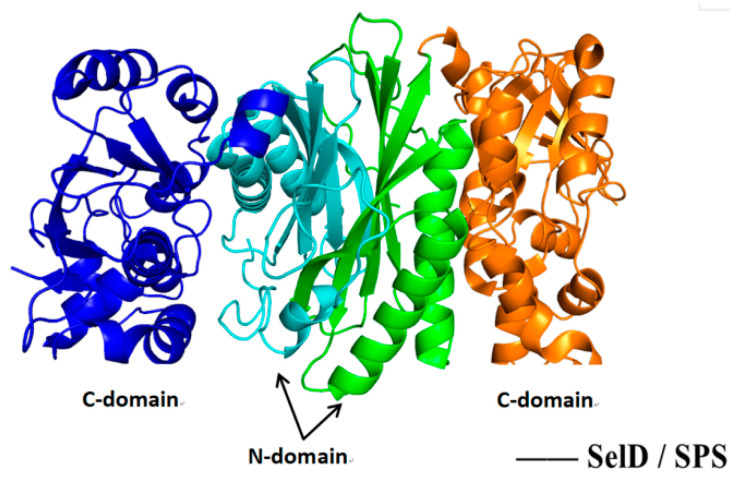
The crystal structure of SelD/SPS in bacteria. Adapted with permission from reference [58].

## Data Availability

Not applicable.

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
