# Peer review of "Mechanisms Affecting the Biosynthesis and Incorporation Rate of Selenocysteine"

_molecules, 2021, doi:10.3390/molecules26237120_

Round 1

Reviewer 1 Report

The review by Wang and coworkers summarizes the pathway of selenocysteine biosynthesis and the factors and elements affecting the incorporation of selenocysteine. This is a useful review that provides a comprehensive view of the mechanisms of the biosynthesis and incorporation of selenocysteine as well as a scientific basis for the industrial fermentation production of selenoproteins. The important references necessary to give an overview of the historical aspects of the mechanistic discussion are also well organized. I have no objection to its publication in Molecules.

Minor issues:

Line 42: “detoxification” is a duplicate.

Figure 2: “0-phosphoseryl-tRNA” should read “O-phosphoseryl-tRNA”.

Line 263: “Yuzuru” should read “Itoh”.

Line 442: “Biotechnology Advances” should read “Current Biotechnology”

Author Response

Dear Reviewer,

Firstly, thank you very much for reviewing my manuscript in your busy schedule and for your recognition of my manuscript. You have provided so specific revision suggestions about four minor errors in your review comments. I do agree with these opinions and will revise them in the new manuscript to be submitted.

Yours sincerely,

Dr. Peng

Reviewer 2 Report

The authors should consider the following suggestions before I can recommend this manuscript for publication.

  1. Abstract: Lines 10-16: It is very hard to follow and understand what the authors are trying to convey. Please use short sentences and simpler English.
  2. I would recommend taking professional help with English for improving the text.
  3. Use short sentences all along the body of the manuscript.
  4. All the figures and flowcharts are very low resolution. I would recommend improving them to make them more reader-friendly.
  5. Are structures of Sel A,B,C, and D are know? If so, then please mention the PDB. Also provide some structural information using figures. This will help in explaining to the reader what you trying to explain.
  6. Lines 105-108: the subheadings are very confusing. Are any sections missing? Please concise your subheadings.
  7. Lines 186-200: Notes sections if very confusing and disturbs the flow of reading. Please change it or remove it.
  8. Lines 94 and 95: references are hypertexted. Please correct them.
  9. Line 215: why is ‘apical’ underlined. Correct it.
  10. Line 262: why is ‘termination codon’ underlined. Correct it.
  11. The authors do not use ‘et al’ whenever a reference is mentioned by name in the manuscript. Please correct it.
  12. Line 262: why is ‘aminoacylate’ underlined. Correct it.
  13. Line 313: use the equation in a separate line. Highlighting it in this way will make it more visible to the reader.
  14. What is ‘nanometer Se’, please explain.
  15. Line 375: please correct the English.
  16. Line 379: Liang et al, please change.
  17. Line 393: are there different kinds of ribosomes that translate these proteins. If so, then please explain.
  18. Line 405-406: correct the English.
  19. Subheading 7.3: consider changing it. It is confusing.

Author Response

Dear Reviewer 2,

Thank you very much for your comments concerning our manuscript entitled “Mechanisms affecting the biosynthesis and incorporation rate of Selenocysteine”. These comments are all extremely valuable and very helpful for revising and improving our paper, as well as guiding important significance to our research. We have carefully studied the comments and made revisions accordingly, which are marked in red in the revised paper. Enclosed please find the point-by-point response to the comments.

We really appreciate the comments from you to improve our work. We also hope that this version is more suitable for publication in this journal.

Thank you very much for your consideration.

Sincerely yours,

Dr. Peng

Point-by-point Response to Reviewer 2:

Reviewer #2

The authors should consider the following suggestions before I can recommend this manuscript for publication.

Response: Thanks for your suggestions. We have carefully revised the manuscript according to you comments, and the point-by-point responses are described as follows.

Point 1: Abstract: Lines 10-16: It is very hard to follow and understand what the authors are trying to convey. Please use short sentences and simpler English.

Response 1: Thanks for your comment and constructive suggestion. The lines introduced that this review provides a comprehensive view of the mechanisms of the biosynthesis and incorporation of selenocysteine as well as a scientific basis for the industrial fermentation production of selenoproteins. To make it more clearly, the abstract was rewritten by using short sentences and simpler English. The revised abstract in the manuscript reads as follows:

Selenocysteine (Sec) is the 21st non-standard proteinogenic amino acid. Due to the particularity of the codon encoding Sec, the selenoprotein synthesis needs to be completed by unique mechanisms in specific biological systems. In this paper, the underlying mechanisms for the biosynthesis and incorporation of Sec into selenoprotein were comprehensively reviewed in five aspects: i) the specific biosynthesis mechanism of Sec and the role of its internal influencing factors (SelA, SelB, SelC, SelD, SPS2 and PSTK); ii) the elements (SECIS, PSL, SPUR and RF) on mRNA and their functional mechanisms; iii) the specificity (either translation termination or translation into Sec) of UGA; iv) the structure-activity relationship and action mechanism of SelA, SelB, SelC and SelD; v) the operating mechanism of two key enzyme systems for inorganic selenium source flow before Sec synthesis. Lastly, the size of the translation initiation interval, other action modes of SECIS and effects of REPS (Repetitive Extragenic Palindromic Sequences) were also discussed in order to improve the incorporation efficiency of Sec and provide scientific basis for the large-scale industrial fermentation production of selenoprotein.

Point 2: I would recommend taking professional help with English for improving the text.

Response 2: Thanks for your suggestion. The manuscript has been carefully checked and revised with the help of some native English speakers.

Point 3: Use short sentences all along the body of the manuscript.

Response 3: Thanks for your comment. The manuscript has been carefully checked with the help of some native English speakers. And we have updated the manuscript with short sentences where necessary.

Point 4: All the figures and flowcharts are very low resolution. I would recommend improving them to make them more reader-friendly.

Response 4: Thanks for your suggestion. The resolution of all the figures and flowcharts in the revised manuscript was increased to over 1200 dpi.

Point 5: Are structures of Sel A, B, C, and D are known? If so, then please mention the PDB. Also provide some structural information using figures. This will help in explaining to the reader what you trying to explain.

Response 5: Thanks for your constructive suggestions. The crystal structures of Sel A, B, C, and D have been resolved, which can be retrieved from the PDB under the accession numbers of 3W1I, 4ZU9, 3A3A and 3U0O respectively. We have added these structural figures in the revised manuscript.

Point 6: Lines 105-108: the subheadings are very confusing. Are any sections missing? Please concise your subheadings.

Response 6: Thanks for your comment. The subheadings from line 105 to 108 have been refined.

Point 7: Lines 186-200: Notes sections if very confusing and disturbs the flow of reading. Please change it or remove it.

Response 7: Thanks for your comments. Notes sections from line 186 to 200 are helpful to understand the figures, and the sentences have been simplified.

Point 8: Lines 94 and 95: references are hypertexted. Please correct them.

Response 8: Thanks for your comments. The hypertexted references have been corrected.

Point 9: Line 215: why is ‘apical’ underlined. Correct it.

Response 9: Thanks for your comments. The underline of ‘apical’ has been removed.

Point 10: Line 262: why is ‘termination codon’ underlined. Correct it.

Response 10: Thanks for your comments. The underline of ‘termination codon’ has been removed.

Point 11: The authors do not use ‘et al’ whenever a reference is mentioned by name in the manuscript. Please correct it.

Response 11: Thanks for your comments. We set 10 authors to be displayed in “References” section originally. Now the ‘et al’ has been deleted and the lacking authors in 3 references have been added.

Point 12: Line 298: why is ‘aminoacylate’ underlined. Correct it.

Response 12: Thanks for your comments. The underline of ‘aminoacylate’ has been removed.

Point 13: Line 313: use the equation in a separate line. Highlighting it in this way will make it more visible to the reader.

Response 13: Thanks for your suggestion. The equation in line 313 has been highlighted in a separate line to be more visible.

Point 14: What is ‘nanometer Se’, please explain.

Response 14: Thanks for your comments. ‘nanometer Se’ is ‘nano-selenium’. To keep the context consistent, We have replaced ‘nanometer Se’ with ‘nano-selenium’.

Point 15: Line 375: please correct the English.

Response 15: Thanks for your comments. The sentences in line 375 have been corrected.

Point 16: Line 382: Liang et al, please change.

Response 16: Thanks for your comments. The sentences in line 382 have been corrected.

Point 17: Line 393: are there different kinds of ribosomes that translate these proteins. If so, then please explain.

Response 17: Thanks for your comments. The reference only informs us that some ribosomes can affect the read through rate of Sec, but doesn’t describe which kinds of ribosomes translate these proteins.

Point 18: Line 405-406: correct the English.

Response 18: Thanks for your comments. The sentences in line 405-406 have been corrected.

Point 19: Subheading 7.3: consider changing it. It is confusing.

Response 19: Thanks for your comments. The subheading 7.3 has been updated to be more clear.

Round 2

Reviewer 2 Report

Based on the revisions done by the authors I can now recommend this manuscript for publication.